

# HSMVS: heuristic search for minimum vertex separator on massive graphs

Chuan Luo and Shanyu Guo

School of Software, Beihang University, Beijing, China

## ABSTRACT

In graph theory, the problem of finding minimum vertex separator (MVS) is a classic NP-hard problem, and it plays a key role in a number of important applications in practice. The real-world massive graphs are of very large size, which calls for effective approximate methods, especially heuristic search algorithms. In this article, we present a simple yet effective heuristic search algorithm dubbed HSMVS for solving MVS on real-world massive graphs. Our HSMVS algorithm is developed on the basis of an efficient construction procedure and a simple yet effective vertex-selection heuristic. Experimental results on a large number of real-world massive graphs present that HSMVS is able to find much smaller vertex separators than three effective heuristic search algorithms, indicating the effectiveness of HSMVS. Further empirical analyses confirm the effectiveness of the underlying components in our proposed algorithm.

## INTRODUCTION

In graph theory, there exist a variety of well-known combinatorial optimization problems, which have extensive important real-world applications in practice (*Wang et al., 2017*; *Li et al., 2017a*; *Wang et al., 2018a*; *Li, Li & Yin, 2019*; *Sun et al., 2023*). Many effective algorithms have been proposed for solving these combinatorial optimization problems, and they achieve good performance on academic benchmarks (mainly randomly generated graphs and crafted graphs). Along with the rapid evolution of the Internet, the rapid growth of real-world networks has resulted in more massive graphs. These real-world massive graphs bring new challenges for practical solving, as existing algorithms usually become ineffective when dealing with them (*Cai, 2015*). The appearance of massive graphs urgently calls for efficient algorithms, since efficient algorithms for solving combinatorial optimization problems would bring much benefit in practice.

Given an undirected graph $G = (V, E)$, where each vertex $v_i \in V$ is associated with a positive integer $c_i$ as its cost, and a positive integer $b$ $(1 \le b \le 2/3|V|)$, which stands for the limitation size, a vertex separator $C$ is a subset of $V$, whose removal partitions the remaining collection of vertices (*i.e.*, $V \setminus C$) into two components, such the size of each component (*i.e.*, the number of vertices in each component) is no greater than $b$.

The minimum vertex separator (MVS) problem is to find such a vertex separator with the smallest total cost in the given graph. In theory, the MVS problem, focusing on finding such a vertex separator of the minimum total cost has been proven to be NP-hard

Corresponding author
Chuan Luo,
chuanluophd@outlook.com

(*Bui & Jones, 1992*; *Fukuyama, 2006*). Besides its considerable importance in theory, the MVS problem is of great significance in practice: MVS has a broad range of useful applications in real-world practical applications, *e.g.,* VLSI design, computational biology, parallel processing, and hyper-graph partitioning (*Balas & de Souza, 2005*; *Evrendilek, 2008*; *Biha & Meurs, 2011*; *Kayaaslan et al., 2012*; *Benlic & Hao, 2013*; *Gomes et al., 2023*), and MVS techniques has been utilized to quantify robustness in complex networks and detect network bottlenecks in communication networks (*Montes-Orozco et al., 2022*; *Montes-Orozco et al., 2021*; *Zhang & Shao, 2015*).

Practical algorithms for solving MVS can be mainly categorized into two classes: exact algorithms and heuristic search algorithms. On one hand, most previous works on solving MVS focus on designing and improving exact algorithms (*Balas & de Souza, 2005*; *de Souza & Balas, 2005*; *Biha & Meurs, 2011*; *de Souza & Cavalcante, 2011*; *Althoby, Biha & Sesboüé, 2020*). These exact algorithms are able to solve relatively small-sized graphs efficiently. However, it is well acknowledged that existing exact algorithms become ineffective on solving large-sized graphs, and may fail to return good-quality solutions within reasonable time on handling those large-sized ones (*Benlic & Hao, 2013*) On the other hand, compared to exact algorithms, heuristic search algorithms, mainly local search ones (*Li et al., 2017b*; *Wang et al., 2018b*; *Serna et al., 2021*; *Zhou, Liu & Gao, 2023*), have exhibited their effectiveness on solving large-sized instances in the context of a variety of combinatorial problems, including Boolean satisfiability (SAT) (*Luo, Hoos & Cai, 2020*), maximum satisfiability (MAX-SAT) (*Luo et al., 2015*; *Cai et al., 2016*; *Luo et al., 2017*), set covering (*Wang et al., 2019*; *Wang et al., 2021*; *Luo et al., 2022*), combinatorial test generation (*Luo et al., 2021c*; *Luo et al., 2021a*), minimum vertex cover (*Luo et al., 2019*; *Li et al., 2020*), minimum dominating set (*Wang et al., 2020c*; *Hu et al., 2021a*; *Li et al., 2022*; *Chen et al., 2023*), maximum clique (*Li et al., 2018*; *Chu et al., 2020*; *Wang et al., 2020a*; *Chu et al., 2023*), graph coloring (*Wang et al., 2020b*), clique partitioning (*Hu et al., 2021b*), virtual machine provisioning (*Luo et al., 2020*; *Luo et al., 2021b*), and container reallocation (*Qiao et al., 2021*).

Also, there exist several heuristic search algorithms for MVS (*Benlic & Hao, 2013*; *Zhang & Shao, 2015*; *Benlic, Epitropakis & Burke, 2017*). However, on real-world massive graphs, the performance of such existing heuristic search algorithms for MVS (*Benlic & Hao, 2013*; *Zhang & Shao, 2015*; *Benlic, Epitropakis & Burke, 2017*) degrades significantly, which can be witnessed from our experimental results (Table 1–Table 7). Hence, it is very interesting to design efficient MVS heuristic search algorithms for handling real-world massive graphs.

In this article, we present an efficient MVS heuristic search algorithm named *HSMVS*, which concentrates on only one simple yet effective vertex-selection heuristic. *HSMVS* first utilizes an efficient construction procedure to initialize the solution, and then applies a vertex-selection heuristic to modify the solution. The vertex-selection heuristic combines random walk and the approximate best selection strategy in an effective way to strike a good balance between intensification and diversification. In order to evaluate the effectiveness of our *HSMVS* algorithm, we conduct extensive experiments to empirically compare *HSMVS* against *BLS*,*BLS-RLE* and *New_K-OPT* on a broad range of real-world massive

**Table 1 Results on the graph classes of biological networks and collaboration networks.**

| Graph | HSMVS | | BLS | | New_K-OPT | | BLS-RLE | |
|---|---|---|---|---|---|---|---|---|
| | best (avg.) | time | best (avg.) | time | best (avg.) | time | best (avg.) | time |
| *Graph Class: Biological Networks* | | | | | | | | |
| bio-celegans | 28 (28.0) | 1.7 | 28 (28.2) | 218.3 | 98 (101.5) | 393.5 | 28 (28.0) | 5.6 |
| bio-diseasome | 4 (4.0) | <0.1 | 4 (4.0) | 0.8 | 83 (83.6) | 391.3 | 4 (4.0) | 6.0 |
| bio-dmela | 826 (828.8) | 318.1 | 890 (935.0) | 862.9 | N/A (N/A) | N/A | 839 (847.8) | 528.8 |
| bio-yeast | 38 (38.7) | 272.4 | 40 (41.0) | 202.1 | 258 (266.7) | 798.9 | 39 (40.6) | 390.9 |
| *Graph Class: Collaboration Networks* | | | | | | | | |
| ca-AstroPh | 1,746 (1,759.9) | 431.8 | 2,726 (2,846.5) | 829.0 | N/A (N/A) | N/A | 2,281 (2,287.6) | 614.1 |
| ca-CSphd | 2 (2.0) | 119.3 | 3 (3.0) | 142.5 | N/A (N/A) | N/A | 3 (3.4) | 246.3 |
| ca-CondMat | 1,134 (1,141.1) | 497.1 | 2,336 (2,424.2) | 675.8 | N/A (N/A) | N/A | 1,582 (1,612.2) | 685.2 |
| ca-Erdos992 | 113 (114.9) | 365.6 | 116 (117.9) | 330.8 | N/A (N/A) | N/A | 120 (123.0) | 348.6 |
| ca-GrQc | 152 (154.2) | 406.2 | 162 (211.1) | 636.7 | N/A (N/A) | N/A | 160 (164.8) | 475.6 |
| ca-HepPh | 738 (743.5) | 407.0 | 1,319 (1,365.5) | 649.4 | N/A (N/A) | N/A | 982 (1,002.9) | 605.2 |
| ca-MathSciNet | 10,216 (10,256.0) | 957.6 | 31,474 (32,670.4) | 775.1 | N/A (N/A) | N/A | 15,053 (17,356.2) | 994.9 |
| ca-citeseer | 4,389 (4,585.1) | 922.6 | 14,669 (15,292.1) | 903.5 | N/A (N/A) | N/A | 10,715 (10,762.3) | 924.083 |
| ca-coauthors-dblp | 36,801 (37,532.2) | 859.9 | 76,018 (77,685.1) | 908.9 | N/A (N/A) | N/A | 82,385 (253,513.8) | 817.3 |
| ca-dblp-2010 | 6,696 (7,190.5) | 962.3 | 16,868 (17,393.6) | 994.6 | N/A (N/A) | N/A | 12,354 (12,794.9) | 977.4 |
| ca-dblp-2012 | 10,817 (11,631.9) | 961.4 | 24,972 (28,280.7) | 991.7 | N/A (N/A) | N/A | 21,666 (22,224.3) | 984.8 |
| ca-hollywood-2009 | 157,694 (164,713.1) | 992.8 | 204,984 (222,974.6) | 999.2 | N/A (N/A) | N/A | N/A (N/A) | N/A |
| ca-netscience | 3 (3.0) | 0.1 | 3 (3.0) | 0.3 | 54 (55.8) | 394.6 | 3 (3.0) | 2.9 |

graphs. The experimental results present that *HSMVS* is able to find better solutions than *BLS*, *BLS-RLE*, and *New_K-OPT* on a large number of graphs. Also, we conduct further empirical evaluations to confirm the effectiveness of the random walk component and the approximate best selection component underlying the *HSMVS* algorithm.

The remainder of this article is organized as follows. In 'Related Work', we give a brief review on MVS solving from the perspectives of both theory and practice. In 'Preliminaries', we provide necessary definitions, concepts and notations. In 'Heuristic Search Framework for Solving MVS', we present a simple heuristic search framework for solving MVS. In 'The *HSMVS* Algorithm', we propose a new heuristic search algorithm called *HSMVS*, and introduce the construction procedure and the modification heuristic of the algorithm in detail. In 'Experiments', extensive experiments comparing *HSMVS* against an effective breakout local search algorithm *BLS* and it's optimized version *BLS-RLE* and an improved K-OPT local search algorithm *New_K-OPT* on a wide range of real-world massive graphs are presented. In 'Discussions', we conduct more empirical evaluations to study the effectiveness of the underlying components in the *HSMVS* algorithm. In 'Conclusions and Future Work', we give the conclusions of this article and list the future work.

## RELATED WORK

Minimum vertex separator is an important NP-hard combinatorial optimization problem in graph theory, and attracts more attentions from academia. Furthermore, this problem is

**Table 2  Results on the graph classes of Facebook networks and infrastructure networks.**

| Graph | HSMVS | | BLS | | New_K-OPT | | BLS-RLE | |
|---|---|---|---|---|---|---|---|---|
| | best (avg.) | time | best (avg.) | time | best (avg.) | time | best (avg.) | time |
| *Graph Class: Facebook Networks* | | | | | | | | |
| socfb-A-anon | 879,079 (907,015.5) | 1,000.0 | N/A (N/A) | N/A | N/A (N/A) | N/A | N/A (N/A) | N/A |
| socfb-B-anon | 775,418 (804,764.4) | 1,000.0 | N/A (N/A) | N/A | N/A (N/A) | N/A | N/A (N/A) | N/A |
| socfb-Berkeley13 | 7,050 (7,061.3) | 565.6 | 6,621 (7,025.7) | 161.5 | N/A (N/A) | N/A | 6,575 (7,040.8) | 559.1 |
| socfb-CMU | 1,805 (1,849.1) | 451.7 | 1,701 (1,834.0) | 39.7 | N/A (N/A) | N/A | 1,684 (1,837.3) | 464.8 |
| socfb-Duke14 | 2,976 (3,089.0) | 308.5 | 2,784 (2,964.4) | 176.0 | N/A (N/A) | N/A | 2,733 (2,867.4) | 531.6 |
| socfb-Indiana | 10,217 (10,418.6) | 552.0 | 9,584 (10,081.9) | 210.3 | N/A (N/A) | N/A | 9,546 (9,747.0) | 472.6 |
| socfb-MIT | 2,027 (2,066.2) | 389.1 | 1,908 (1,975.7) | 269.4 | N/A (N/A) | N/A | 1,864 (1,909.9) | 403.2 |
| socfb-OR | 9,108 (9,146.5) | 549.7 | 9,410 (10,061.6) | 780.3 | N/A (N/A) | N/A | 8,901 (9,145.5) | 815.9 |
| socfb-Penn94 | 12,439 (12,547.7) | 423.0 | 11,927 (13,063.4) | 585.6 | N/A (N/A) | N/A | 11,744 (12,125.1) | 737.1 |
| socfb-Stanford3 | 3,475 (3,547.6) | 420.1 | 3,154 (3,262.8) | 14.2 | N/A (N/A) | N/A | 3,117 (3,147.1) | 628.5 |
| socfb-Texas84 | 12,884 (13,150.2) | 684.1 | 12,070 (12,712.6) | 120.6 | N/A (N/A) | N/A | 12,023 (12,691.8) | 657.4 |
| socfb-UCLA | 6,339 (6,426.8) | 354.8 | 5,930 (6,252.7) | 221.4 | N/A (N/A) | N/A | 5,868 (6,253.5) | 614.1 |
| socfb-UCSB37 | 4,373 (4,380.7) | 582.7 | 4,199 (4,512.5) | 249.5 | N/A (N/A) | N/A | 4,185 (4,258.5) | 537.6 |
| socfb-UConn | 5,515 (5,601.3) | 507.2 | 5,212 (5,681.3) | 341.0 | N/A (N/A) | N/A | 5,186 (5,342.1) | 560.1 |
| socfb-UF | 12,032 (12,226.6) | 550.5 | 11,314 (11,624.3) | 429.1 | N/A (N/A) | N/A | 11,279 (11,554.6) | 694.3 |
| socfb-UIllinois | 10,148 (10,316.1) | 516.2 | 9,531 (10,177.1) | 360.3 | N/A (N/A) | N/A | 9,489 (9,707.2) | 619.4 |
| socfb-Wisconsin87 | 7,787 (7,856.0) | 499.7 | 7,320 (7,481.9) | 327.4 | N/A (N/A) | N/A | 7,277 (7,599.4) | 500.0 |
| socfb-uci-uni | 18,002,372 (18,028,784.7) | 999.9 | N/A (N/A) | N/A | N/A (N/A) | N/A | N/A (N/A) | N/A |
| *Graph Class: Infrastructure Networks* | | | | | | | | |
| inf-power | 10 (13.6) | 330.8 | 257 (312.2) | 410.7 | N/A (N/A) | N/A | 8 (9.8) | 87.5 |
| inf-road-usa | 9,196,531 (9,199,294.7) | 1,000.0 | N/A (N/A) | N/A | N/A (N/A) | N/A | N/A (N/A) | N/A |
| inf-roadNet-CA | 443,008 (477,729.7) | 1,000.0 | N/A (N/A) | N/A | N/A (N/A) | N/A | N/A (N/A) | N/A |
| inf-roadNet-PA | 32,481 (32,815.2) | 999.7 | 218,854 (219,358.9) | 1,000.0 | N/A (N/A) | N/A | N/A (N/A) | N/A |

becoming increasingly important because it has shown to have real-world applications in practice. Thus, there exist a number of works which are devoted to solving MVS in either theory or practice. In this section, we give a brief review on MVS solving, and discuss MVS algorithms from the perspectives of both theory and practice.

## Theoretical algorithms

Because MVS has proven to be NP-hard (*Bui & Jones, 1992*; *Fukuyama, 2006*), it seems impossible to design exact algorithms with the complexity of polynomial time. Thus, most theoretical works on MVS focused on designing approximation algorithms, which aims at improving the approximation ratio for this NP-hard combinatorial optimization problem. *Leighton & Rao (1999)* presented an approximation algorithm for MVS, which is based on linear programming, and the algorithm gives an approximation ratio of $O(\log n)$ for MVS. Then, *Feige, Hajiaghayi & Lee (2008)* developed an approximation algorithm for MVS, which is based on novel linear and semidefinite program relaxations, and obtained the approximation ratio of $O(\log \sqrt{opt})$, where *opt* is the size of an optimal vertex separator.

**Table 3  Results on the graph classes of interaction networks, recommendation networks, Retweet networks and scientific computing.**

| Graph | HSMVS | | BLS | | New_K-OPT | | BLS-RLE | |
|---|---|---|---|---|---|---|---|---|
| | best (avg.) | time | best (avg.) | time | best (avg.) | time | best (avg.) | time |
| *Graph Class: Interaction Networks* | | | | | | | | |
| ia-email-EU | 363 (381.3) | 393.2 | 200 (214.4) | 574.8 | N/A (N/A) | N/A | 188 (195.3) | 531.0 |
| ia-email-univ | 150 (154.6) | 44.9 | 151 (156.3) | 39.9 | 417 (420.6) | 664.8 | 149 (149.7) | 261.0 |
| ia-enron-large | 657 (677.4) | 592.1 | 1,688 (2,018.1) | 864.3 | N/A (N/A) | N/A | 792 (820.2) | 388.3 |
| ia-enron-only | 18 (18.0) | <0.1 | 18 (18.0) | 0.1 | 30 (32.1) | 312.1 | 18 (18.0) | 3.7 |
| ia-fb-messages | 223 (225.1) | 507.0 | 215 (216.9) | 416.3 | 446 (452.4) | 800.5 | 216 (225.6) | 302.3 |
| ia-infect-dublin | 26 (26.0) | 12.5 | 26 (26.0) | 0.4 | 120 (125.0) | 473.6 | 26 (26.0) | 4.4 |
| ia-infect-hyper | 47 (47.0) | 0.1 | 47 (47.0) | <0.1 | 52 (52.0) | 62.1 | 47 (47.0) | 13.1 |
| ia-reality | 43 (45.2) | 432.8 | 25 (26.1) | 377.4 | N/A (N/A) | N/A | 28 (29.1) | 519.6 |
| ia-wiki-Talk | 4,080 (4,163.9) | 723.3 | 5,931 (6,095.9) | 902.1 | N/A (N/A) | N/A | 3,882 (3,926.2) | 608.9 |
| *Graph Class: Recommendation Networks* | | | | | | | | |
| rec-amazon | 937 (989.8) | 992.8 | 4,532 (4,622.4) | 100.7 | N/A (N/A) | N/A | 506 (569.9) | 980.9 |
| *Graph Class: Retweet Networks* | | | | | | | | |
| rt-retweet-crawl | 27,470 (27,780.4) | 993.0 | 46,949 (48,199.5) | 870.5 | N/A (N/A) | N/A | 35,531 (38,339.5) | 989.5 |
| rt-retweet | 4 (4.0) | <0.1 | 4 (4.0) | 0.1 | 13 (14.2) | 459.3 | 4 (4.0) | 2.8 |
| rt-twitter-copen | 15 (15.0) | 0.3 | 15 (15.0) | 24.3 | 139 (141.7) | 535.0 | 15 (15.4) | 232.4 |
| *Graph Class: Scientific Computing* | | | | | | | | |
| sc-ldoor | 12,271 (15,369.9) | 997.9 | 78,416 (82,497.3) | 998.1 | N/A (N/A) | N/A | N/A (N/A) | N/A |
| sc-msdoor | 2,647 (3,939.7) | 956.1 | 19,715 (21,021.4) | 976.4 | N/A (N/A) | N/A | 2,325 (3,906.5) | 979.8 |
| sc-nasasrb | 314 (618.6) | 502.1 | 270 (433.1) | 199.4 | N/A (N/A) | N/A | 270 (405.0) | 103.9 |
| sc-pkustk11 | 1,509 (1,778.0) | 513.1 | 1,380 (2,578.4) | 564.6 | N/A (N/A) | N/A | 1,344 (1,355.4) | 240.5 |
| sc-pkustk13 | 1,249 (1,548.6) | 601.0 | 1,254 (4,256.9) | 782.5 | N/A (N/A) | N/A | 1,137 (1,195.8) | 761.9 |
| sc-pwtk | 834 (2,192.2) | 666.7 | 720 (3,885.9) | 441.4 | N/A (N/A) | N/A | 720 (1,121.1) | 371.8 |
| sc-shipsec1 | 2,226 (3,821.0) | 941.5 | 12,379 (15,933.3) | 955.0 | N/A (N/A) | N/A | 2,313 (3,165.5) | 870.6 |
| sc-shipsec5 | 3,407 (4,172.8) | 767.6 | 13,373 (18,961.9) | 946.6 | N/A (N/A) | N/A | 2,602 (3,865.0) | 886.7 |

## Practical algorithms

Even though a number of great contributions have been made on the theoretical analysis of MVS solving, the performance of theoretical algorithms for MVS is still unsatisfactory in practice. As MVS has important applications in real-world situations, such as VLSI design, computational biology, etc (*Balas & de Souza, 2005*; *Biha & Meurs, 2011*; *Benlic & Hao, 2013*; *Zhang & Shao, 2015*; *Dagdeviren, Akram & Farzan, 2019*; *Furini et al., 2022*), a number of practical algorithms for MVS have been proposed.

As mentioned in the 'Introduction', practical algorithms for MVS can be classified into two categories: exact algorithms and heuristic search algorithms. Exact algorithms are guaranteed to prove optimal solutions, but they may fail to return good-quality solutions within reasonable time on solving large-sized instances (*Benlic & Hao, 2013*). Heuristic search algorithms could not prove optimality for the solutions they find, but they are able to seek out good-quality solutions for large-sized instances efficiently.

Most previous works on practical MVS solving focus on designing and improving exact algorithms. In *de Souza & Balas (2005)*, developed a branch-and-cut algorithm for MVS,

**Table 4  Results on the graph classes of social networks and technological networks.**

| Graph | HSMVS | | BLS | | New_K-OPT | | BLS-RLE | |
|---|---|---|---|---|---|---|---|---|
| | best (avg.) | time | best (avg.) | time | best (avg.) | time | best (avg.) | time |
| *Graph Class: Social Networks* | | | | | | | | |
| soc-BlogCatalog | 7,866 (8,020.8) | 769.5 | 8,792 (10,162.4) | 934.2 | N/A (N/A) | N/A | 5,189 (5,549.6) | 733.4 |
| soc-FourSquare | 7,806 (12,731.9) | 986.4 | 45,709 (46,519.7) | 978.3 | N/A (N/A) | N/A | 28,282 (148,884.4) | 865.2 |
| soc-LiveMocha | 18,332 (18,362.2) | 569.5 | 18,797 (20,718.9) | 956.2 | N/A (N/A) | N/A | 16,908 (17,660.2) | 708.0 |
| soc-brightkite | 3,249 (3,366.2) | 745.0 | 5,299 (5,451.3) | 773.1 | N/A (N/A) | N/A | 2,832 (3,098.6) | 835.3 |
| soc-buzznet | 8,507 (9,670.0) | 644.4 | 11,337 (11,686.9) | 952.2 | N/A (N/A) | N/A | 5,012 (5,361.3) | 834.2 |
| soc-delicious | 11,397 (15,247.0) | 991.8 | 25,308 (28,000.5) | 948.4 | N/A (N/A) | N/A | 14,035 (17,291.2) | 988.7 |
| soc-digg | 30,202 (30,424.3) | 981.0 | 43,319 (45,162.7) | 989.4 | N/A (N/A) | N/A | 36,411 (43,237.7) | 985.5 |
| soc-dolphins | 6 (6.0) | <0.1 | 6 (6.0) | <0.1 | 16 (16.0) | 41.8 | 6 (6.0) | 2.2 |
| soc-douban | 5,171 (5,178.7) | 702.5 | 6,588 (6,799.8) | 696.6 | N/A (N/A) | N/A | 5,110 (5,170.0) | 447.5 |
| soc-epinions | 1,319 (1,332.4) | 550.3 | 2,114 (2,304.0) | 584.4 | N/A (N/A) | N/A | 1,401 (1,442.5) | 628.5 |
| soc-flickr | 19,308 (19,483.3) | 977.3 | 37,663 (40,705.5) | 995.2 | N/A (N/A) | N/A | 33,546 (56,574.6) | 968.9 |
| soc-flixster | 543,056 (553,120.9) | 1,000.0 | N/A (N/A) | N/A | N/A (N/A) | N/A | N/A (N/A) | N/A |
| soc-gowalla | 5,355 (6,569.1) | 898.7 | 11,956 (16,307.1) | 926.8 | N/A (N/A) | N/A | 6,881 (8,838.6) | 970.0 |
| soc-karate | 4 (4.0) | <0.1 | 4 (4.0) | <0.1 | 4 (4.0) | 17.0 | 4 (4.0) | 18.2 |
| soc-lastfm | 44,117 (44,392.8) | 996.9 | 49,735 (52,096.3) | 860.5 | N/A (N/A) | N/A | N/A (N/A) | N/A |
| soc-livejournal | 1,617,709 (1,619,315.0) | 1,000.0 | N/A (N/A) | N/A | N/A (N/A) | N/A | N/A (N/A) | N/A |
| soc-orkut | 1,289,460 (1,296,974.2) | 1,000.0 | N/A (N/A) | N/A | N/A (N/A) | N/A | N/A (N/A) | N/A |
| soc-pokec | 357,172 (388,602.4) | 1,000.0 | 641,157 (644,721.7) | 1,000.0 | N/A (N/A) | N/A | N/A (N/A) | N/A |
| soc-slashdot | 5,962 (5,990.6) | 785.5 | 7,618 (7,660.1) | 921.3 | N/A (N/A) | N/A | 5,904 (5,932.8) | 793.1 |
| soc-twitter-follows | 2,571 (2,673.0) | 501.6 | 1,797 (1,886.3) | 903.7 | N/A (N/A) | N/A | 1,131 (1,155.5) | 584.3 |
| soc-wiki-Vote | 52 (52.0) | 8.0 | 51 (51.2) | 179.6 | 265 (270.0) | 526.5 | 51 (51.4) | 11.1 |
| soc-youtube-snap | 31,524 (32,040.8) | 997.7 | 61,245 (71,035.8) | 907.4 | N/A (N/A) | N/A | N/A (N/A) | N/A |
| soc-youtube | 18,342 (19,615.1) | 968.1 | 37,746 (40,604.2) | 954.3 | N/A (N/A) | N/A | 28,588 (32,091.3) | 994.6 |
| *Graph Class: Technological Networks* | | | | | | | | |
| tech-RL-caida | 3,435 (3,773.5) | 983.7 | 14,522 (15,974.7) | 770.6 | N/A (N/A) | N/A | 4,349 (5,107.7) | 978.8 |
| tech-WHOIS | 209 (222.1) | 494.7 | 348 (440.1) | 380.8 | N/A (N/A) | N/A | 269 (277.2) | 466.7 |
| tech-as-caida2007 | 149 (157.7) | 720.4 | 615 (714.6) | 533.1 | N/A (N/A) | N/A | 269 (276.1) | 515.1 |
| tech-as-skitter | 335,256 (352,827.6) | 1,000.0 | 650,515 (686,833.9) | 11.6 | N/A (N/A) | N/A | N/A (N/A) | N/A |
| tech-internet-as | 248 (276.1) | 733.3 | 892 (1,183.4) | 753.5 | N/A (N/A) | N/A | 476 (491.3) | 649.0 |
| tech-p2p-gnutella | 6,226 (6,230.8) | 793.7 | 8,716 (8,883.0) | 851.0 | N/A (N/A) | N/A | 6,243 (6,255.3) | 716.8 |
| tech-routers-rf | 97 (98.2) | 284.9 | 95 (97.1) | 462.7 | N/A (N/A) | N/A | 92 (94.9) | 504.3 |

which is based on the mixed integer programming formulation (*Balas & de Souza, 2005*). After that, *Biha & Meurs (2011)* designed an exact algorithm for MVS, on the basis of new classes of valid inequalities for the associated polyhedron. Further, *de Souza & Cavalcante (2011)* proposed a hybrid algorithm, which is built on a Lagrangian relaxation framework. Recently, *Althoby, Biha & Sesboüé (2020)* introduced a practical method which combines branch-and-bound procedure, linear programming technique and greedy algorithm.

In the context of MVS solving by heuristic search, *Benlic & Hao (2013)* developed the first local search algorithm called *BLS* for solving MVS. In order to improve the

**Table 5  Results on the graph class of temporal reachability networks.**

| Graph | HSMVS | | BLS | | New_K-OPT | | BLS-RLE | |
|---|---|---|---|---|---|---|---|---|
| | best (avg.) | time | best (avg.) | time | best (avg.) | time | best (avg.) | time |
| *Graph Class: Temporal Reachability Networks* | | | | | | | | |
| scc_enron-only | 66 (67.3) | <0.1 | 66 (67.8) | <0.1 | 66 (66.0) | 24.7 | 66 (67.5) | 39.7 |
| scc_fb-forum | 132 (150.6) | 529.9 | 142 (157.8) | 271.0 | 137 (211.2) | 497.8 | 149 (155.3) | 523.8 |
| scc_fb-messages | 524 (600.1) | 472.9 | 474 (485.1) | 101.7 | N/A (N/A) | N/A | 475 (490.7) | 575.3 |
| scc_infect-dublin | 0 (0.0) | 285.1 | 0 (0.0) | 78.7 | N/A (N/A) | N/A | 0 (0) | 0.6 |
| scc_infect-hyper | 85 (85.0) | 0.4 | 85 (85.0) | <0.1 | 85 (86.7) | 242.0 | 85 (88.4) | 49.3 |
| scc_reality | 3,009 (3,201.6) | 635.2 | 1,170 (1,223.5) | 282.0 | N/A (N/A) | N/A | 1,173 (1,175.6) | 544.1 |
| scc_retweet-crawl | 4 (5.2) | 476.4 | 0 (0.0) | 23.4 | N/A (N/A) | N/A | 0 (0.0) | 11.7 |
| scc_retweet | 14 (27.6) | 572.8 | 1 (11.8) | 118.2 | N/A (N/A) | N/A | 1 (57.0) | 79.4 |
| scc_rt_alwefaq | 0 (0.1) | 120.3 | 0 (0.0) | 0.4 | N/A (N/A) | N/A | 0 (0.0) | <0.1 |
| scc_rt_assad | 0 (0.0) | <0.1 | 0 (0.0) | 0.2 | N/A (N/A) | N/A | 0 (0.0) | <0.1 |
| scc_rt_bahrain | 0 (0.0) | <0.1 | 0 (0.0) | 0.3 | N/A (N/A) | N/A | 0 (0.0) | <0.1 |
| scc_rt_barackobama | 0 (0.0) | 0.2 | 0 (0.0) | 0.3 | N/A (N/A) | N/A | 0 (0.0) | <0.1 |
| scc_rt_damascus | 0 (0.0) | <0.1 | 0 (0.0) | 0.3 | N/A (N/A) | N/A | 0 (0.0) | <0.1 |
| scc_rt_dash | 0 (0.0) | <0.1 | 0 (0.0) | 0.3 | N/A (N/A) | N/A | 0 (0.0) | <0.1 |
| scc_rt_gmanews | 0 (9.3) | 37.3 | 0 (0.0) | 0.3 | N/A (N/A) | N/A | 0 (0.0) | <0.1 |
| scc_rt_gop | 0 (0.0) | <0.1 | 0 (0.0) | 0.3 | N/A (N/A) | N/A | 0 (0.0) | <0.1 |
| scc_rt_http | 0 (0.0) | <0.1 | 0 (0.0) | 0.3 | N/A (N/A) | N/A | 0 (0.0) | <0.1 |
| scc_rt_israel | 0 (0.0) | <0.1 | 0 (0.0) | 0.3 | N/A (N/A) | N/A | 0 (0.0) | <0.1 |
| scc_rt_justinbieber | 0 (1.4) | 154.9 | 0 (0.0) | 0.4 | N/A (N/A) | N/A | 0 (0.0) | <0.1 |
| scc_rt_ksa | 0 (0.0) | <0.1 | 0 (0.0) | 0.3 | N/A (N/A) | N/A | 0 (0.0) | <0.1 |
| scc_rt_lebanon | 0 (0.0) | <0.1 | 0 (0.0) | 0.2 | N/A (N/A) | N/A | 0 (0.0) | <0.1 |
| scc_rt_libya | 0 (0.0) | <0.1 | 0 (0.0) | 0.4 | N/A (N/A) | N/A | 0 (0.0) | <0.1 |
| scc_rt_lolgop | 0 (1.3) | 559.7 | 0 (0.0) | 0.3 | N/A (N/A) | N/A | 0 (0.0) | <0.1 |
| scc_rt_mittromney | 0 (0.0) | <0.1 | 0 (0.0) | 0.3 | N/A (N/A) | N/A | 0 (0.0) | <0.1 |
| scc_rt_obama | 0 (0.0) | <0.1 | 0 (0.0) | 0.2 | N/A (N/A) | N/A | 0 (0.0) | <0.1 |
| scc_rt_occupy | 0 (0.0) | <0.1 | 0 (0.0) | 0.3 | N/A (N/A) | N/A | 0 (0.0) | <0.1 |
| scc_rt_occupywallstnyc | 0 (4.8) | 315.0 | 0 (0.0) | 0.3 | N/A (N/A) | N/A | 0 (0.0) | <0.1 |
| scc_rt_oman | 0 (0.0) | <0.1 | 0 (0.0) | 0.3 | N/A (N/A) | N/A | 0 (0.0) | <0.1 |
| scc_rt_onedirection | 0 (0.0) | <0.1 | 0 (0.0) | 0.3 | N/A (N/A) | N/A | 0 (0.0) | <0.1 |
| scc_rt_p2 | 0 (0.0) | <0.1 | 0 (0.0) | 0.3 | N/A (N/A) | N/A | 0 (0.0) | <0.1 |
| scc_rt_qatif | 0 (0.0) | <0.1 | 0 (0.0) | 0.3 | N/A (N/A) | N/A | 0 (0.0) | <0.1 |
| scc_rt_saudi | 0 (0.0) | <0.1 | 0 (0.0) | 0.3 | N/A (N/A) | N/A | 0 (0.0) | <0.1 |
| scc_rt_tcot | 0 (0.0) | <0.1 | 0 (0.0) | 0.3 | N/A (N/A) | N/A | 0 (0.0) | <0.1 |
| scc_rt_tlot | 0 (0.0) | <0.1 | 0 (0.0) | 0.3 | N/A (N/A) | N/A | 0 (0.0) | <0.1 |
| scc_rt_uae | 0 (0.0) | <0.1 | 0 (0.0) | 0.3 | N/A (N/A) | N/A | 0 (0.0) | <0.1 |
| scc_rt_voteonedirection | 0 (0.0) | <0.1 | 0 (0.0) | 0.1 | N/A (N/A) | N/A | 0 (0.0) | <0.1 |
| scc_twitter-copen | 486 (579.4) | 537.0 | 400 (417.0) | 590.5 | N/A (N/A) | N/A | 390 (420.5) | 431.9 |

**Table 6  Results on the graph class of web graphs.**

| Graph | HSMVS | | BLS | | New_K-OPT | | BLS-RLE | |
|---|---|---|---|---|---|---|---|---|
| | best (avg.) | time | best (avg.) | time | best (avg.) | time | best (avg.) | time |
| *Graph Class: Web Graphs* | | | | | | | | |
| web-BerkStan | 37 (40.3) | 538.6 | 55 (490.0) | 331.1 | N/A (N/A) | N/A | 66 (70.9) | 521.8 |
| web-arabic-2005 | 11 (17.4) | 641.8 | 3,508 (3,769.3) | 905.0 | N/A (N/A) | N/A | 184 (197.8) | 651.9 |
| web-edu | 2 (2.0) | 116.6 | 2 (2.0) | 0.5 | N/A (N/A) | N/A | 2 (2.0) | 8.3 |
| web-google | 4 (4.0) | 0.6 | 4 (4.0) | 150.9 | 123 (131.7) | 797.6 | 4 (4.0) | 192.0 |
| web-indochina-2004 | 8 (9.8) | 463.0 | 31 (43.8) | 348.9 | N/A (N/A) | N/A | 21 (24.8) | 522.9 |
| web-it-2004 | 6 (7.9) | 616.6 | 10,909 (11,622.3) | 867.4 | N/A (N/A) | N/A | 1,474 (2,014.8) | 975.3 |
| web-polblogs | 24 (24.0) | 0.2 | 24 (24.0) | 67.1 | 131 (137.5) | 489.6 | 24 (24.0) | 4.1 |
| web-sk-2005 | 29 (40.3) | 767.9 | 3,504 (3,859.5) | 842.8 | N/A (N/A) | N/A | 85 (89.9) | 836.0 |
| web-spam | 475 (479.2) | 530.9 | 458 (564.1) | 564.4 | N/A (N/A) | N/A | 458 (464.8) | 505.4 |
| web-uk-2005 | 1 (1.0) | 281.7 | 1 (1.0) | 33.0 | N/A (N/A) | N/A | 1 (1.0) | 218.7 |
| web-webbase-2001 | 3 (5.6) | 541.5 | 22 (25.7) | 443.0 | N/A (N/A) | N/A | 26 (30.5) | 483.0 |
| web-wikipedia2009 | 328,301 (329,736.9) | 1,000.0 | N/A (N/A) | N/A | N/A (N/A) | N/A | N/A (N/A) | N/A |

**Table 7  Overall results on all real-world massive graphs.**

| Graph class | #graph | HSMVS | | BLS | | New_K-OPT | | BLS-RLE | |
|---|---|---|---|---|---|---|---|---|---|
| | | #best (#avg.) | time | #best (#avg.) | time | #best (#avg.) | time | #best (#avg.) | time |
| Biological Networks | 4 | 4 (4) | 148.1 | 2 (1) | 321.0 | 0 (0) | 486.2 | 2 (2) | 234.6 |
| Collaboration Networks | 13 | 13 (13) | 606.4 | 1 (1) | 679.8 | 0 (0) | 394.6 | 1 (1) | 657.2 |
| Facebook Networks | 18 | 3 (3) | 575.3 | 0 (2) | 285.8 | 0 (0) | N/A | 15 (13) | 533.6 |
| Infrastructure Networks | 4 | 3 (3) | 832.6 | 0 (0) | 705.3 | 0 (0) | N/A | 1 (1) | 258.4 |
| Interaction Networks | 9 | 4 (4) | 300.7 | 5 (5) | 352.8 | 0 (0) | 407.9 | 6 (6) | 407.9 |
| Recommendation Networks | 1 | 0 (0) | 992.8 | 0 (0) | 100.7 | 0 (0) | N/A | 1 (1) | 981.0 |
| Retweet Networks | 3 | 3 (3) | 331.1 | 2 (2) | 298.3 | 0 (0) | 497.1 | 2 (1) | 407.9 |
| Scientific Computing | 8 | 2 (1) | 743.3 | 2 (0) | 733.0 | 0 (0) | N/A | 6 (7) | 631.7 |
| Social Networks | 23 | 15 (15) | 742.4 | 3 (3) | 764.9 | 1 (1) | 195.1 | 10 (9) | 641.3 |
| Technological Networks | 7 | 6 (6) | 715.8 | 0 (0) | 625.3 | 0 (0) | N/A | 1 (1) | 494.4 |
| Temporal Reachability Networks | 37 | 32 (26) | 127.0 | 35 (34) | 39.8 | 2 (1) | 237.5 | 33 (31) | 264.4 |
| Web Graphs | 12 | 11 (11) | 458.3 | 5 (4) | 414.0 | 0 (0) | 643.6 | 5 (5) | 283.7 |
| Total | 139 | 96 (89) | 468.2 | 55 (52) | 392.8 | 3 (2) | 392.3 | 83 (78) | 421.5 |

performance, *BLS* incorporates several sophisticated heuristics (including a greedy hill-climbing component, an adaptive perturbation mechanism, a hashing function and a jumping-magnitude determining component), which introduce six instance-dependent parameters. There exists an improved version of *BLS*, which is called *BLS-RLE* (*Benlic, Epitropakis & Burke, 2017*). *BLS-RLE* introduces an effective parameter control mechanism that draws upon ideas from reinforcement learning theory to reach an interdependent decision. According to the computational results reported in the literature (*Benlic & Hao, 2013*; *Benlic, Epitropakis & Burke, 2017*), *BLS* is able to handle graphs with up to

3,000 vertices and runs much faster than a number of high-performance exact algorithms and *BLS-RLE* exhibits its effectiveness in solving the MVS problem. Besides, *Zhang et al. (2015)* proposed an improved K-OPT local search algorithm named *New_K-OPT*. The experimental results reported in the literature (*Zhang et al., 2015*) show that *New_K-OPT* exhibits relatively better performance compared to variable neighborhood search, simulated annealing and Relax-and-Cut (*de Souza & Cavalcante, 2011*) on a number of graphs.

## PRELIMINARIES

In this section, we give necessary backgrounds of the minimum vertex separator (MVS) problem. An undirected graph $G = (V, E)$ consists of a set of vertices $V$ and a set of edges $E \subseteq V \times V$, where each edge $e$ is a pair of two different vertices in $V$. For an edge $e = (u, v)$, we say that vertices $u$ and $v$ are the endpoints of edge $e$. Two different vertices are neighbors if and only if they both appear at the same edge. We use the notation $N(v) = \{u | (u, v) \in E \text{ and } u \neq v\}$ to denote the set of $v$'s all neighboring vertices. The degree of a vertex $v$ is denoted as $deg(v) = |N(v)|$.

Given an undirected graph, where each vertex is associate with a positive integer as its cost, and a limitation size, a vertex separator is a subset of vertices, whose removal divides the remaining vertices into two disjoint components (*i.e.,* there is no edge connected those two components), subject to the size of each component (*i.e.,* the number of vertices in each component) smaller than the limitation size. In this article, we address the problem of finding such a vertex separator as small total cost as possible.

More formally, given an undirected graph $G = (V, E)$ with a cost $c_i$ corresponding to each vertex $v_i \in V$ and a positive integer $b$ ($1 \leq b \leq |V|$) denoting the limitation size, the minimum vertex separator (MVS) problem is to find a partition which divides $V$ into three disjoint subsets $A$, $B$ and $C$, such that (i) $A$ and $B$ are non-empty; (ii) there is no edge $(v_i, v_j) \in E$ with $v_i \in A$ and $v_j \in B$; (iii) $|A| \leq b$ and $|B| \leq b$, where $0 \leq b \leq 2/3|V|$; and (iv) $\sum_{v_j \in C} c_j$ is minimized.

The vertex separator $C$ is feasible when the the first three constraints (i, ii and iii) are satisfied, and is optimal when all constraints are satisfied. In theory, the MVS problem with $0 \leq b \leq 2/3|V|$ has been proven to be NP-hard (*Feige & Mahdian, 2006*; *Bui & Jones, 1992*; *Fukuyama, 2006*). Since the empirical study on solving MVS (*Benlic & Hao, 2013*) demonstrates that it is computationally difficult to solve the MVS problem with $b = \frac{1.05|V|}{2}$, in this work we follow this setting, and in our major experiments (as presented in 'Experiments') $b$ is set to $\frac{1.05|V|}{2}$ accordingly. Moreover, we would like to note that, in 'Discussions' we conduct empirical evaluations with $b = 0.6$, so as to study the performance of our proposed algorithm under different values of $b$.

The concept of solution is very important in heuristic search algorithms.

In the MVS problem (where $b$ denotes the limitation size), a partition $s = \{A, B, C\}$, which divides the vertex set $V$ into three disjoint subsets and guarantees that the sizes of $A$ and $B$ are not greater than $b$, is called a solution. The cost of a is solution $s$, denoted as $cost(s)$, is the sum of the cost $c_j$ of each $v_j \in C$ (*i.e.,* $cost(s) = \sum_{v_j \in C} c_j$). Obviously, the less the value of $cost(s)$ is, the better the quality of solution $s$ is. Hence, the MVS problem aims to find a solution $s$ of minimum cost.

---

**Algorithm 1** Heuristic Search Framework for MVS

---

**Input**: Graph $G$, limitation size $b$;

**Output**: A solution $s^*$;

1:   $s \leftarrow Construct\_Solution(G, b)$, $s^* \leftarrow s$;

2:   **while** terminating criterion is not reached **do**

3:       $s \leftarrow Modify\_Solution(s, b)$;

4:       **if** $cost(s) < cost(s^*)$ **then** $s^* \leftarrow s$;

5:   **end while**

6:   **return** $s^*$;

---

## HEURISTIC SEARCH FRAMEWORK FOR SOLVING MVS

As described in 'Introduction', heuristic search, especially local search, is a popular paradigm and recently has shown effectiveness on a variety of NP-hard combinatorial problems. The basic idea of local search is that, it firstly constructs a solution as the initial solution, and then iteratively applies heuristics, which modify the resulting solution, to improve the solution quality (which is the cost of the solution, as defined in 'Preliminaries'). Obviously, because combinatorial problems are rather different from each other in nature, it is difficult to solve a specific problem by directly applying heuristics designed for other problems. Therefore, it is a challenge to design an effective heuristic search algorithm for solving a combinatorial problem.

*BLS* has introduced the first local search framework for MVS (*Benlic & Hao, 2013*). This framework is composed of several heuristics and thus is relatively complex. In this section, we introduce a simple heuristic search framework for MVS, in order to demonstrate the most essential parts in heuristic search algorithms for solving MVS.

The basic heuristic search framework for MVS is outlined in Algorithm 1 as described as follows. In the beginning, heuristic search calls the function *Construct_Solution* to generate a solution $s$ as the initial solution, and the best solution $s^*$ is initialized as $s$ (line 1). After the initialization, heuristic search conducts the search stage iteratively until the terminating criterion is reached (lines 2–5). In each search step, heuristic search modifies solution $s$ by employing the function *Modify_Solution* (line 3); whenever a better solution with a smaller cost is found, the best solution $s^*$ is updated accordingly (line 4). After the search stage, the resulting solution $s^*$ is reported as the final solution (line 6).

## THE *HSMVS* ALGORITHM

On the basis of the simple heuristic search framework in the preceding section, we develop a new heuristic search algorithm called *HSMVS* for solving MVS. In this section, we present the whole *HSMVS* algorithm in detail.

---

**Algorithm 2** The Function *Construct_Solution*

---

**Input**: Graph $G = (V, E)$, limitation size $b$;

**Output**: A solution $s = \{A, B, C\}$;

  1:   Initialized three vertex set $A, B, C$ to $\phi$;
  2:   **foreach** *vertex* $v \in V$ **do**
  3:      **if** *with probability p* **then**
  4:         **if** $|A| < b$ **then** put $v$ into set $A$;
  5:         **else if** $|B| < b$ **then** put $v$ into set $B$;
  6:         **else then** put $v$ into set $C$;
  7:      **else**
  8:         **if** $|B| < b$ **then** put $v$ into set $B$;
  9:         **else if** $|A| < b$ **then** put $v$ into set $A$;
10:         **else then** put $v$ into set $C$;
11:      **end if**
12:   **end foreach**
13:   **foreach** *vertex* $v \in B$ **do**
14:      **if** $N(v) \cap A \neq \phi$ **then** move $v$ from $B$ to $C$;
15:   **end foreach**
16:   **return** $s = \{A, B, C\}$;

---

According to the pseudo-code in Algorithm 1, it is clear that the functions *Construct_Solution* and *Modify_Solution* are the most crucial parts in this framework. Thus, we specify these two functions in our *HSMVS* algorithm.

In order to build an effective heuristic search algorithm, our *HSMVS* algorithm utilizes an efficient heuristic function named *Construct_Solution* to construct the initial solution. We outline the pseudo-code of the function *Construct_Solution* in Algorithm 2. We note that the construction procedure consists of an extending stage and a fixing stage, which are described as below.

## The construction procedure

**The extending stage:** In the beginning, three vertex sets $A$, $B$ and $C$ are initialized as $\varnothing$ (line 1). Then, for each vertex $v \in V$, the function puts $v$ into one of these three sets according to the following rules.

- With probability $p$, if $|A| < b$, the function puts $v$ into $A$; if $|A| \geq b$ and $|B| < b$, the function puts $v$ into $B$; if $|A| \geq b$ and $|B| \geq b$, the function puts $v$ into $C$ (lines 3–6).
- Otherwise (with probability $1 - p$), if $|B| < b$, the function puts $v$ into $B$; if $|B| \geq b$ and $|A| < b$, the function puts $v$ into $A$; if $|B| \geq b$ and $|A| \geq b$, the function puts $v$ into $C$ (lines 7–11).

**The fixing stage:** According to the rules in the extending stage, there might be a number of edges that connect some vertices in set $A$ and their neighboring vertices in set $B$, which makes the resulting solution $\{A, B, C\}$ infeasible. To construct a feasible solution, the function tries to move some vertices from set $B$ to set $C$. For each vertex $v \in B$, the function

---

**Algorithm 3** The Function *Modify_Solution*

---

**Input**: A source solution $s = \{A, B, C\}$, limitation size $b$;

**Output**: A modified solution $s' = \{A', B', C'\}$;

1: **if** *with a probability wp* **then**
2:     $v \leftarrow$ a random vertex in set $C$;
3:     $X \leftarrow$ a random set from $\{A, B\}$;
4:     **if** $X == A$ **then** *move(v, A, B)*;
5:     **if** $X == B$ **then** *move(v, B, A)*;
6: **else**
7:     $v_A \leftarrow$ a random vertex from set $C$;
8:     **for** $i \leftarrow 1$ **to** $t - 1$ **do**
9:         $r_A \leftarrow$ a random vertex from set $C$;
10:        **if** $score_A(r_A) > score_A(v_A)$ **then** $v_A \leftarrow r_A$;
11:    **end for**
12:    $v_B \leftarrow$ a random vertex from set $C$;
13:    **for** $i \leftarrow 1$ **to** $t - 1$ **do**
14:        $r_B \leftarrow$ a random vertex from set $C$;
15:        **if** $score_B(r_B) > score_B(v_B)$ **then** $v_B \leftarrow r_B$;
16:    **end for**
17:    **if** $|A| == b$ **then** *move(v_B, B, A)*;
18:    **else then** *move(v_A, A, B)*;
19:    **if** $score_A(v_A) > score_B(v_B)$ **then** *move(v_A, A, B)*;
20:    **else then** *move(v_B, B, A)*;
21: **end if**
22: $A' \leftarrow A$, $B' \leftarrow B$, $C' \leftarrow C$;
23: **return** $s' = \{A', B', C'\}$;

---

checks whether there are neighboring vertices of $v$ in set $A$; if this is the case, the function moves vertex $v$ into set $C$ in order to resolve the contradiction (lines 13–15). Finally, the function returns $s = \{A, B, C\}$ as the solution.

**Example 1** *To make readers better understand our proposed algorithm, we present an example here to demonstrate how our construction procedure constructs an initial solution in a high-level sense.* Figure 1 *illustrates an example graph, which has eight vertices and nine edges, and we assume that the cost of each vertex is 1, indicating that the costs of all vertices are the same. For the example graph in* Fig. 1 *, given the limitation size $b$ of 4 (i.e., $b = 4$), once the extending stage is completed, an infeasible solution could be generated. For instance, assuming the constructed infeasible solution is comprised of $A = \{4, 6, 7\}$, $B = \{0, 1, 2, 3, 5\}$ and $C = \varnothing$, since some vertices in set $B$ (i.e., vertices 1, 2 and 5) have neighboring vertices in $A$, during the fixing stage, those vertices of 1, 2 and 5 would be moved from set $B$ to set $C$, resulting in a feasible solution of $A = \{4, 6, 7\}$, $B = \{0, 3\}$ and $C = \{1, 2, 5\}$.*

As stated in the literature (*Cai, 2015*), it is important to design low time-complexity function to generate the initial solution for massive graphs, because high time-complexity

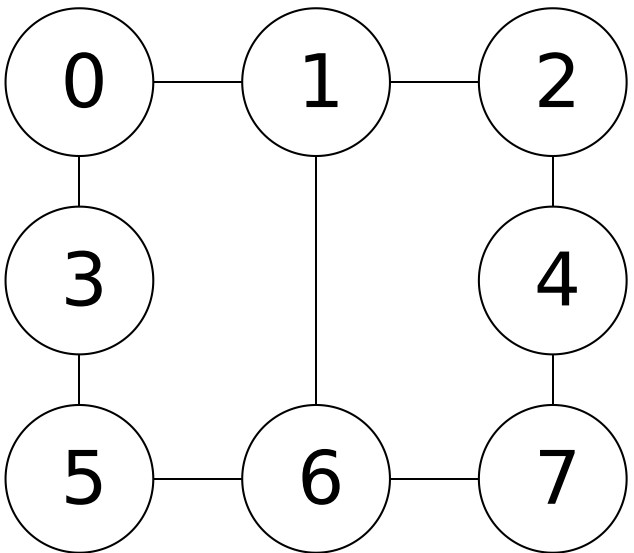

**Figure 1    An example graph.**

construction procedure would make the algorithms inefficient. According to the pseudo-code in Algorithm 2, we could easily derive the following lemma (Lemma 1).

**Lemma 1**   *The time-complexity of the function Construct_Solution in Algorithm 2 is* $O(max\{|V|, |E|\})$.

**Proof** The function Construct_Solution can be divided into two main stages: the extending stage (lines 2–12) and the fixing stage (lines 13–15). Each part consists of a loop.For the extending stage, it is clear that the time-complexity is $O(|V|)$, as each vertex $v \in V$ is visited.For the fixing stage, the time-complexity could be calculated as $\sum_{v \in B}|N(v)| = \sum_{v \in B}deg(v)$. Because set $B$ is a subset of set $V$, we have $\sum_{v \in B}deg(v) \leq \sum_{v \in V}deg(v) = 2 \times |E|$. Thus, the time-complexity of the second loop is $O(|E|)$. Therefore, according to above analysis, we are able to derive that the time-complexity of the whole function *Construct_Solution* is $O(max\{|V|, |E|\})$.

In fact, most real-world massive graphs are sparse ones (*Barabási & Albert, 1999*; *Eubank et al., 2004*; *Cai, 2015*). Thus, in most cases, the complexity of our function *Construct_Solution* is usually lower than $O(|V|^2)$, which indicates that our construction procedure is practical for a large number of real-world massive graphs.

## The modification heuristic

The modification heuristic also plays a critical role in the *HSMVS* algorithm. An important issue of designing an effective modification heuristic is to strike a good balance between intensification and diversification (*Li & Huang, 2005*). Inspired by the success of two-mode heuristic search algorithms in Boolean satisfiability solving (*Balint & Fröhlich, 2010*; *Li & Li, 2012*; *Cai & Su, 2013*; *Luo, Su & Cai, 2014*), we propose an effective two-mode modification heuristic named *Modify_Solution* in the context of MVS solving.

Essentially, the heuristic *Modify_Solution* modifies the current solution by moving a vertex $v$ from set $C$ to the target set $X \in \{A, B\}$ and resolving the contradictions by moving to set $C$ those vertices, which are $v$'s neighbors and are currently in the opposite set $Y$ ($Y = \{A, B\} \setminus X$). Clearly, the most important issue of the heuristic is to decide the moving vertex $v$ and the target set $X$.

Before describing the details of *Modify_Solution*, we introduce the basic operation in the heuristic. The operation $move(v, X, Y)$, where $v \in C$, $X \in \{A, B\}$, $Y = \{A, B\} \setminus X$, works as follows. It first moves vertex $v$ from set $C$ to set $X$. Then, for each vertex $w \in Y$, it checks whether $w \in N(v)$; if this is the case, it moves $w$ from set $Y$ to set $C$ to keep the solution legal. We also introduce two evaluating properties $score_A$ and $score_B$, which are important metrics for evaluating the priority of vertices in set $C$. The formal definitions of $score_A$ and $score_B$ are given as follows (Definitions 1 and 2).

**Definition 1** *Given a solution $s = \{A, B, C\}$, for each vertex $v \in C$, the property $score_A(v)$ is defined as the decrement in the $cost(s)$ after executing the operation $move(v, A, B)$.*

**Definition 2** *Given a solution $s = \{A, B, C\}$, for each vertex $v \in C$, the property $score_B(v)$ is defined as the decrement in the $cost(s)$ after executing the operation $move(v, B, A)$.*

Given a vertex $v \in C$, the evaluation properties $score_A$ and $score_B$ represent the benefit through performing the operations $move(v, A, B)$ and $move(v, B, A)$, respectively. Also, performing an operation with larger value of $score_A$ or $score_B$ would reduce the value of cost to the largest extent. Therefore, it is advisable to select and conduct an operation with large value of $score_A$ or $score_B$.

**Example 2** *For the example graph in Fig. 1, we assume that each vertex has the same cost of 1, and the current solution is $s = \{A, B, C\}$, where $A = \{4, 6, 7\}$, $B = \{0, 3\}$, $C = \{1, 2, 5\}$. For vertex $2 (2 \in C)$, if we move vertex 2 from set $C$ to set $A$, because vertex 2 has no neighboring vertex in set $B$, the decrement in the $cost(s)$ after executing the operation move $(2, A, B)$ is 1, so the $score_A(2)$ is 1. If we move vertex 2 from set $C$ to set $B$, since vertex 4 is the neighboring vertex of 2, and vertex $4 (4 \in A)$ should be moved from set $A$ to set $C$, then the decrement in the $cost(s)$ after executing the operation move $(2, B, A)$ is 0 (i.e., $score_B(2)$ is 0). After comparing $score_A(2)$ and $score_B(2)$, we can decide the suitable set to which vertex 2 should be moved.*

These properties play important roles in the reconstruction of solutions and reduction of the cost.

We present the pseudo-code of the whole heuristic *Modify_Solution* in Algorithm 3, and describe it in detail. Our heuristic *Modify_Solution* switches between two modes, *i.e.,* the random mode and the greedy mode, in order to strike a good balance between intensification and diversification. The function *Modify_Solution* activates which mode depending on a probability $wp$. With the probability $wp$, *Modify_Solution* works in the random mode (lines 1–5); otherwise (with the probability $1 - wp$), *Modify_Solution* works in the greedy mode (lines 6–19). The procedures of the random mode and the greedy mode are described as follows.

**The random mode:** In this mode, the heuristic employs the random walk component to strengthen diversification. The random walk component first randomly selects a vertex

$v$ from set $C$, and then randomly picks a target set $X$ from $\{A, B\}$. If set $X$ is set $A$, then the heuristic performs the operation $move(v, A, B)$; otherwise (set $X$ is set $B$), the heuristic executes the operation $move(v, B, A)$.

**The greedy mode:** In this mode, the heuristic applies the approximate best selection component to contribute to intensification, inspired by the success of Best from Multiple Selections (BMS) in the context of minimum vertex cover (*Cai, 2015*). The approximate best selection component first chooses $t$ vertices from set $C$, and among these $t$ vertices selects the vertex with the greatest $score_A$, denoted as $v_A$ (lines 7–11). Then, the heuristic also chooses $t$ vertices from set $C$, and among these $t$ vertices selects the vertex with the greatest $score_B$, denoted as $v_B$ (lines 12–16). Finally, the heuristic checks whether $score_A(v_A)$ is greater than $score_B(v_B)$; if this is the case, it executes the operation $move(v_A, A, B)$; otherwise ($score_A(v_A)$ is not greater than $score_B(v_B)$), it executes the operation $move(v_B, B, A)$.

Finally, our heuristic *Modify_Solution* denotes the resulting sets $A$, $B$, $C$ as sets $A'$, $B'$ and $C'$, respectively, and then returns $s' = \{A', B', C'\}$ as the resulting solution.

**Example 3** *In the greedy mode, we firstly calculate the values of $score_A$ and $score_B$ for each vertex in set $C$. Figure 2 shows the comparison of a solution before and after the movement. From Fig. 2, we can obtain $score_A(1) = 0$, $score_B(1) = 0$, $score_A(2) = 1$, $score_B(2) = 0$, $score_A(5) = 0$, and $score_B(5) = 0$. Since the vertex 2 is with the greatest $score_A$, and also the greatest among all the value of $score_A$ and $score_B$, our heuristic chooses vertex 2 and the set $A$, and then performs the operation $move(2, C, A)$.*

**Remark:** We note that the solution $s'$ returned by the heuristic *Modify_Solution* might be infeasible. If this is the case, the algorithm would first rollback the resulting solution $s' = \{A', B', C'\}$ to $s = \{A, B, C\}$, and then randomly moves a vertex from set $A$ to set $C$ (or moves a vertex from set $B$ to set $C$).

# EXPERIMENTS

In order to show the effectiveness of our *HSMVS* algorithm, we compare *HSMVS* against an effective breakout local search algorithm *BLS* and its optimized version *BLS-RLE* and an improved K-OPT local search algorithm *New_K-OPT* on a broad range of real-world massive graphs. In this section, we first introduce the benchmarks, the competitors and the experimental setup of our experiments. Then we report the comparative results.

## The benchmarks

We evaluate *HSMVS* on all 139 graphs collected in a public and standard graph I benchmark (https://lcs.ios.ac.cn/~caisw/graphs.html), which is originally collected from Network Repository (*Rossi & Ahmed, 2015a*; *Rossi & Ahmed, 2015b*) and consists of a broad range of real-world massive simple undirected graphs. Most of these graphs are encoded from real-world applications. In practice, these real-world massive graphs have been utilized in testing practical algorithms for well-known NP-hard combinatorial optimization problems in graph theory, including minimum vertex cover (*Luo et al., 2019*; *Li et al., 2020*), minimum dominating set (*Chen et al., 2023*), maximum clique (*Rossi et al., 2014*) and graph coloring (*Rossi & Ahmed, 2014*).

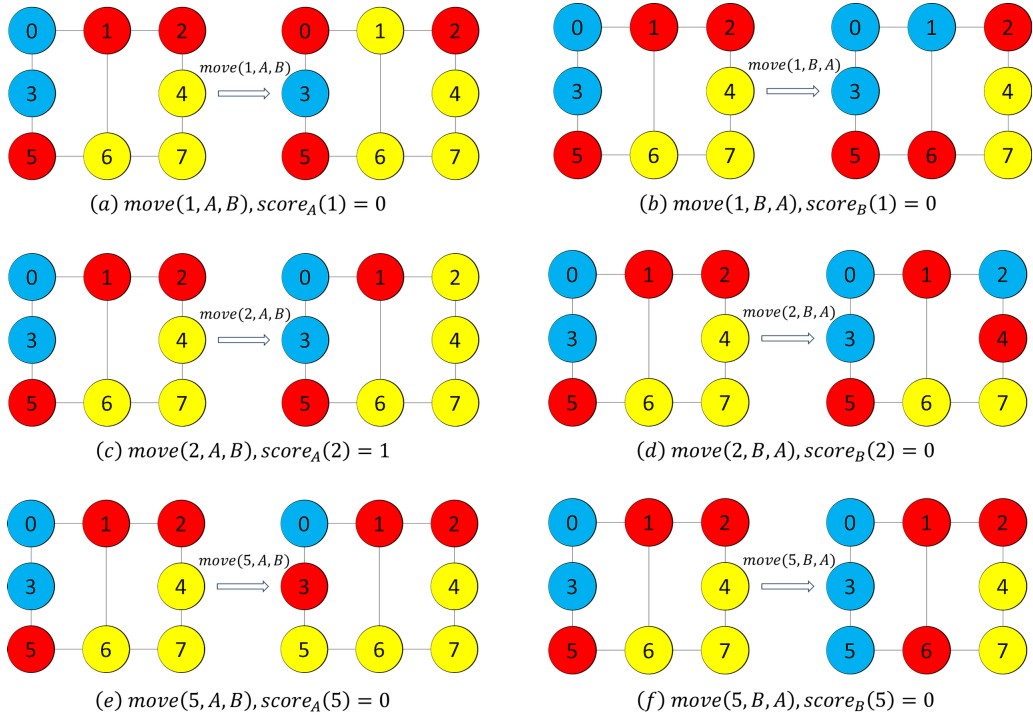

**Figure 2 Example figures to demonstrate movement.** We note that the vertices in red color, yellow color and blue color are the vertices in subsets $A$, $B$ and $C$, respectively.

The graphs tested in our experiments contain a variety of real-world networks, and can be classified into 12 categories, including biological networks, collaboration networks, Facebook networks, infrastructure network, interaction networks, recommendation networks, Retweet networks, scientific computing, social networks, technological networks, temporal reachability networks and web graphs. For these graphs evaluated in our experiments, all the vertices are given unit weights, and $b = \lfloor \frac{1.05|V|}{2} \rfloor$ recalling that $b$ can be regarded as the limitation size and firstly introduced in Section 'Preliminaries'. These benchmarking settings are suggested by the literature (*Benlic & Hao, 2013*).

**The competitors**

Our *HSMVS* algorithm is compared against an effective breakout local search solver *BLS* (*Benlic & Hao, 2013*), an improved K-OPT local search solver *New_K-OPT* (*Zhang & Shao, 2015*), and an optimized version of *BLS*, called *BLS-RLE* (*Benlic, Epitropakis & Burke, 2017*).

- The *BLS* solver (*Benlic & Hao, 2013*) is the first local search solver for solving the MVS problem, and it achieves effectiveness in solving MVS instances. According to the experiments in the literature (*Benlic & Hao, 2013*), *BLS* performs significantly better than a number of high-performance exact solvers (*de Souza & Balas, 2005*; *Biha & Meurs, 2011*). As reported in the literature (*Hager & Hungerford, 2015*), *BLS* exhibits its effectiveness in solving MVS on random graphs.

[1]For each solver run, the corresponding solver would report a final solution. For a solver on a graph, since each solver performs 10 runs, then there would be 10 reported solutions in total, and the best solution is the solution with the smallest cost among all 10 solutions. The best solution quality among all 10 runs is the cost of the best solution.

- The *New_K-OPT* solver (*Zhang & Shao, 2015*) is a high-performance, improved K-OPT local search solver for solving the MVS problem. The experimental results reported in the literature (*Zhang & Shao, 2015*) show that *New_K-OPT* exhibits relatively better performance compared to several methods, such as variable neighborhood search, simulated annealing and Relax-and-Cut (*de Souza & Cavalcante, 2011*), on a number of graphs.

- The *BLS-RLE* solver (*Benlic, Epitropakis & Burke, 2017*) is an enhancement of *BLS*. The *BLS-RLE* solver introduces a new parameter control mechanism, which is designed on the basis of the reinforcement learning theory. As claimed by its authors, this new parameter control mechanism could help the *BLS-RLE* solver better escape from the local optimum situation. According to the experimental results demonstrated in the literature (*Benlic, Epitropakis & Burke, 2017*), *BLS-RLE* performs much better than *BLS* on various types of graphs.

## Experimental setup

Our *HSMVS* algorithm is implemented in the programming language C++. In our experiments, for *HSMVS*, the parameter $p$ is set to 0.5, as the initialization should be uniformly random; the parameter $wp$ is set to 0.05 and the parameter $t$ is set to 20 according to preliminary experiments. The local search competitor *BLS* is an open-source solver and can be downloaded online (http://www.info.univ-angers.fr/pub/hao/BLSVSP/Code/BLS_VSP.cpp). The *BLS* solver is implemented in the programming language C++. For *BLS*, we adopt the parameter settings which are reported in the literature (*Benlic & Hao, 2013*). The *BLS* solver is implemented in the programming language C++. For *BLS-RLE*, its implementation is publicly available online. (http://www.epitropakis.co.uk/BLS-RLE/) The *BLS-RLE* is implemented in the programming language C++, and it is evaluated using the configuration settings that are utilized in the literature (*Benlic, Epitropakis & Burke, 2017*). The source codes of the improved K-OPT local search competitor *New_K-OPT* is kindly provided by its author. The *New_K-OPT* solver is implemented in the programming language C++. For *New_K-OPT*, we adopt the algorithmic settings which are reported in the literature (*Zhang & Shao, 2015*). In order to make the empirical comparison fair, all these three algorithms *HSMVS*, *BLS*, *BLS-RLE* and *New_K-OPT* are statically complied by the compiler g++ with the option '-O3'.

All the experiments are carried out on a number of workstations equipped with Intel Xeon E7-8830 2.13 GHz CPU, 24MB L3 cache and 1.0TB RAM under the operating system CentOS 7.0.1406. In our experiments, each solver performs 10 runs on each graph. The cutoff time of each run performed by each solver is set to 1,000 s.

For each graph, we report the best solution quality found by each solver among all 10 runs, denoted by 'best'[1], the average solution quality over all 10 runs, denoted by 'avg.', and the average run time of reporting the best solution in each run, denoted by 'time'. If a solver fails to report solutions on a graph within the cutoff time among all 10 runs, we mark 'N/A' for 'best', 'avg.' and 'time' for the related solver on the related graph.

Furthermore, for each solver on each graph class, we report the number of graphs where the solver finds the best solution quality among all competing solvers in the related

experiment, denoted by '#best', the number of graphs where the solver finds the best average solution quality among all competing solvers in the related experiment, denoted by '#avg.', and the average time of reporting the best solution in each run, denoted by 'time'. If a solver fails to report solutions on all graphs in a graph class, we mark 'N/A' for 'time' for the related solver on the related graph class. The number of graphs in each graph class is indicated in the column '#graph'.

This form of demonstrating experimental results is inspired by the rules of well-known SAT competitions (http://www.satcompetition.org/) and MAX-SAT evaluations (http://www.maxsat.udl.cat/).

## Experimental results

In this subsection, we first present the experimental results, and then conduct some discussions about the results.

The comparative results of *HSMVS* and its competitors *BLS, BLS-RLE, New_K-OPT* on all real-world massive graphs are reported in Tables 1–7, where Table 1 presents the comparative results on the graph classes of biological networks and collaboration networks, Table 2 presents the comparative results on the graph classes of Facebook networks and infrastructure networks, Table 3 presents the comparative results on the graph classes of interaction networks, recommendation networks, Retweet networks and scientific computing, Table 4 presents the comparative results on the graph classes of social networks and technological networks, Table 5 presents the comparative results on the graph class of temporal reachability networks, Table 6 presents the comparative results on the graph class of web graphs, and Table 7 summarizes the comparative results on all massive real-world graphs.

First we focus on the comparison between *HSMVS* and *BLS*. According to the results reported in Tables 1–7, among all 12 graph classes, it is apparent that our *HSMVS* algorithm performs better than *BLS* on 9 graph classes (*i.e.,* biological networks, collaboration networks, facebook networks, infrastructure networks, retweet networks, scientific computing, social networks, technological networks and web graphs). On the overall performance, according to Table 7, among all 139 real-world massive graphs, our *HSMVS* algorithm finds the best solution quality for 96 of them, while *BLS* does that for only 55 of them; *HSMVS* finds the best average solution quality for 89 of them, while this figure for *BLS* is only 52.

Then we concentrate on the evaluation between *HSMVS* and *New_K-OPT*. According to the results reported in Tables 1–7, it is clear that *HSMVS* significantly outperforms *New_K-OPT* on all 12 graph classes. On the overall performance, seen from Table 7, *HSMVS* gives the best solution quality for 96 of them, while this figure for *New_K-OPT* is only 3; *HSMVS* finds the best average solution quality for 89 of them, while this figure for *New_K-OPT* is only 2.

Finally, we analyze the comparison between *HSMVS* and *BLS-RLE*. According to the results reported in Tables 1–7, our *HSMVS* algorithm performs better than *BLS-RLE* on 7 graph classes (*i.e.,* biological networks, collaboration networks, infrastructure networks, retweet networks, social networks, technological networks, and web graphs). On the overall

performance, according to Table 7, among all 139 real-world massive graphs, our *HSMVS* algorithm finds the best solution quality for 96 of them, while *BLS-RLE* does that for only 83 of them; *HSMVS* finds the best average solution quality for 89 of them, while this figure for *BLS* is only 78.

**Remark:** The experimental results on a broad range of real-world massive graphs in Tables 1–7 present that *HSMVS* generally performs much better than the effective breakout local search competitor *BLS*, the improved K-OPT local search competitor *New_K-OPT*, and the enhanced version of *BLS* named *BLS-RLE*, on a large number of real-world massive graphs, indicating that *HSMVS* shows its superiority on solving real-world massive graphs.

## DISCUSSIONS

In this section, we conduct empirical evaluations to further discuss the effectiveness of *HSMVS*. In particular, we first perform ablation studies to demonstrate the effectiveness of algorithmic components (*i.e.,* the approximate best selection component and the random walk component) underlying *HSMVS*. Then, we analyze the performance of *HSMVS* on different limitation size. Finally, we discuss the advantage of *HSMVS* when compared to its competitors.

### Effectiveness of algorithmic components underlying *HSMVS*

According to the description of the *HSMVS* algorithm, it is obvious that the approximate best selection component in the greedy mode and the random walk component in the random mode are the key parts. In order to show the effectiveness of these two components, we develop three alternative versions of *HSMVS*, which are all modified from *HSMVS* and are described as follows.

- *HSMVS_alt1:* This version uses the strict best selection component instead of the approximate best selection component. *HSMVS_alt1* differs from *HSMVS* in lines 7–16 in Algorithm 3: in lines 7–11, *HSMVS_alt1* greedily selects the variable with the greatest $score_A$ from set $C$, denoted $v_A$; in lines 12–16, *HSMVS_alt2* greedily selects the variable with the greatest $score_B$ from set $C$, denoted as $v_B$.
- *HSMVS_alt2:* This version uses the random selection component instead of the approximate best selection component. *HSMVS_alt2* also differs from *HSMVS* in lines 7–16 in Algorithm 3: in lines 7–11, *HSMVS_alt2* randomly selects a variable from set $C$, denoted as $v_A$; in lines 12–16, *HSMVS_alt2* randomly selects a variable from set $C$, denoted as $v_B$. In another word, *HSMVS_alt2* could be considered as a specific version of *HSMVS* with parameter $t = 1$.
- *HSMVS_alt3:* This version does not utilize the random walk component, *i.e.,* working without the random mode (deleting lines 1–5 in Algorithm 3). In another word, *HSMVS_alt3* could be considered as a specific version of *HSMVS* with parameter $wp = 0$.

Then, we conduct extensive empirical evaluations to compare *HSMVS* with its three alternative versions on the all 139 real-world massive graphs. The experimental setup used in this comparison is the same one used in 'Experiments'. To make the evaluation fair, all

**Table 8  Overall results of *HSMVS* and its three alternative versions on all real-world massive graphs.**

| Graph Class | #graph | HSMVS | | HSMVS_alt1 | | HSMVS_alt1 | | HSMVS_alt3 | |
|---|---|---|---|---|---|---|---|---|---|
| | | #best (#avg.) | time | #best (#avg.) | time | #best (#avg.) | time | #best (#avg.) | time |
| Total | 139 | 85 (83) | 468.2 | 64 (56) | 530.3 | 42 (35) | 364.7 | 71 (67) | 478.0 |

these alternative versions are also implemented in C++, and are statically compiled by g++ with the option '-O3'. Furthermore, the parameters settings used in these three alternative versions are the same as in *HSMVS*.

Table 8 reports the related empirical results of comparing the *HSMVS* algorithm with all its alternative versions (*i.e., HSMVS_alt1, HSMVS_alt2* and *HSMVS_alt3*) on all 139 real-world massive graphs. As can be seen from Table 8, it is clear that *HSMVS* stands out as the general best solver in this comparison. Particularly, *HSMVS* performs much better than all its alternative versions in terms of both the best solution quality and the average solution quality. Among 139 total real-world massive graphs, *HSMVS* finds the best solution quality for 85 of them, while this figure is only 64, 42 and 71 for *HSMVS_alt1, HSMVS_alt2* and *HSMVS_alt3*, respectively; *HSMVS* finds the best average solution quality for 83 of them, while this figure is only 56, 35 and 67 for *HSMVS_alt1, HSMVS_alt2* and *HSMVS_alt3*, respectively.

**Remark:** The empirical results presented in Table 8 show that *HSMVS* generally performs better than all its alternative versions and thus is the general best algorithm on the real-world massive graphs, which confirms the effectiveness of the approximate best component and the random walk component.

### Experiment results on different limitation size

In this subsection, we conduct empirical evaluations to assess the performance of *HSMVS* on different limitation size. In particular, compared to the setting of limitation size ($b = \lfloor \frac{1.05|V|}{2} \rfloor$) that is adopted in Section 'Experiments', here we set the limitation size to $b = \lfloor 0.6|V| \rfloor$. Also, in this subsection we conduct empirical evaluations on 12 selected graphs, where we randomly select a graph from each graph class. Table 9 reports the comparative results of *HSMVS* and its competitors on 12 selected graphs with $b = \lfloor 0.6|V| \rfloor$, and Table 10 summarizes the overall results on those 12 selected graphs with $b = \lfloor 0.6|V| \rfloor$. As can be observed from Tables 9 and 10, our *HSMVS* algorithms still performs generally better than its competitors (*i.e., BLS, BLS-RLE* and *New_K-OPT*). According to Table 10, *HSMVS* gives the best solution quality for 9 of the overall selected graphs, while this figure for *BLS, New_K-OPT* and *BLS-RLE* is 0, 0 and 3, respectively. Also, *HSMVS* finds the best average solution quality for 8 of them, while this figure for *BLS, New_K-OPT* and *BLS-RLE* is 0, 0 and 4, respectively. In summary, *HSMVS* achieves generally better performance than its competitors on a different limitation size (*i.e., $b = \lfloor 0.6|V| \rfloor$*).

### Discussion on the advantage of *HSMVS*

As presented in Tables 1–7, there is no single best algorithm across all classes of graphs. Hence, in this subsection, we aim to discuss the advantage of *HSMVS* when compared to its competitors. Particularly, we analyze the experimental results and the features of graphs,

**Table 9  Results on 12 selected graphs with $b = \lfloor 0.6|V| \rfloor$.**

| Graph | HSMVS | | BLS | | New_K-OPT | | BLS-RLE | |
|---|---|---|---|---|---|---|---|---|
| | best (avg.) | time | best (avg.) | time | best (avg.) | time | best (avg.) | time |
| *Graph Class: Biological Networks* | | | | | | | | |
| bio-dmela | 715 (719.8) | 549.5 | 794 (822.4) | 881.1 | N/A (N/A) | N/A | 728 (737.9 ) | 410.2 |
| *Graph Class: Collaboration Networks* | | | | | | | | |
| ca-CondMat | 1,017 (1,019.9) | 271.6 | 2,090 (2,151.7) | 780.8 | N/A (N/A) | N/A | 1,491 (1,539.5) | 476.9 |
| *Graph Class: Facebook Networks* | | | | | | | | |
| socfb-OR | 8,396 (8,420.7) | 507.9 | 7,995 (8,621.1) | 625.6 | N/A (N/A) | N/A | 7,340 (7,981.0) | 857.2 |
| *Graph Class: Infrastructure Networks* | | | | | | | | |
| inf-power | 9 (12.1) | 396.9 | 80 (239.3) | 597.3 | N/A (N/A) | N/A | 10  (10.0) | 108.2 |
| *Graph Class: Interaction Networks* | | | | | | | | |
| ia-enron-large | 493 (497.2) | 582.2 | 1,531 (1,670.1) | 597.6 | N/A (N/A) | N/A | 586 (600.2) | 628.6 |
| *Graph Class: Recommendation Networks* | | | | | | | | |
| rec-amazon | 820 (859.8) | 988.5 | 4,708 (4,891.1) | 244.1 | N/A (N/A) | N/A | 436 (437.7) | 884.0 |
| *Graph Class: Retweet Networks* | | | | | | | | |
| rt-retweet-crawl | 22,100 (22,169.8) | 956.1 | 47,005 (49,623.6) | 192.2 | N/A (N/A) | N/A | N/A (N/A) | N/A |
| *Graph Class: Scientific Computing* | | | | | | | | |
| sc-ldoor | 8,401 (10,831.2) | 995.6 | 66,452 (73,480.6) | 999.2 | N/A (N/A) | N/A | 80,861 (194,289.5) | 910.5 |
| *Graph Class: Social Networks* | | | | | | | | |
| soc-brightkite | 2,402 (2,648.1) | 766.0 | 3,964 (4,773.8) | 369.1 | N/A (N/A) | N/A | 2,381  (2,484.7 ) | 787.5 |
| *Graph Class: Technological Networks* | | | | | | | | |
| tech-as-caida2007 | 129 (156.2) | 432.8 | 190 (282.5) | 769.5 | N/A (N/A) | N/A | 193 (198) | 551.2 |
| *Graph Class: Temporal Reachability Networks* | | | | | | | | |
| scc_enron-only | 54 (55.5) | <0.1 | 55 (56.6) | <0.1 | N/A (N/A) | N/A | 54 (55.6) | 38.3 |
| *Graph Class: Web Graphs* | | | | | | | | |
| web-BerkStan | 36 (38.2) | 323.0 | 55 (302.9) | 316.3 | N/A (N/A) | N/A | 66 (71.7) | 613.9 |

**Table 10  Overall results on 12 selected graphs with $b = \lfloor 0.6|V| \rfloor$.**

| Graph Class | #graph | HSMVS | | BLS | | New_K-OPT | | BLS-RLE | |
|---|---|---|---|---|---|---|---|---|---|
| | | #best (#avg.) | time | #best (#avg.) | time | #best (#avg.) | time | #best (#avg.) | time |
| Total | 12 | 9 (8) | 468.5 | 0 (0) | 531.1 | 0 (0) | N/A | 3 (4) | 522.2 |

for identifying the characteristics of graphs which *HSMVS* exhibits better effectiveness than its competitors. Figure 3 illustrates the relationship between the practical performance of competing algorithms (including *HSMVS* and its competitors) and the size of graphs (*i.e.,* the number of graphs' vertices). In Fig. 3, the $X$-axis depicts $\ln(|V|)$, where $|V|$ represents the number of vertices, while the $Y$-axis presents $\ln(|avg| + 1)$, where $|avg|$ denotes the corresponding algorithm's obtained average solution quality over all 10 runs. It can be observed that our *HSMVS* algorithm shows competitive performance on graphs with relatively large number of vertices. As discussed in 'The *HSMVS* Algorithm', our *HSMVS* algorithm strikes a good balance between intensification and diversification. In this way, when handling graphs with relatively large numbers of vertices, compared to its

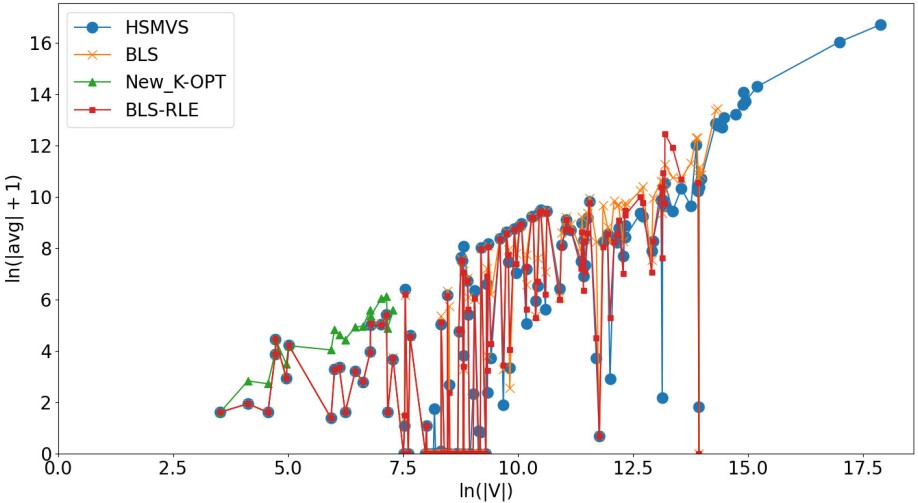

**Figure 3** Practical performance of competing algorithms (including *HSMVS* and its competitors) on graphs with different sizes.

competitors, our *HSMVS* algorithm is able to explore a broader solution space in a shorter time, resulting in an advantage on larger-scale graphs.

## CONCLUSIONS AND FUTURE WORK

In this work, we present an effective MVS heuristic search algorithm called *HSMVS*, which introduces an efficient construction procedure and an effective vertex-selection heuristic. To demonstrate the effectiveness of our *HSMVS* algorithm, we conduct extensive experiments to compare *HSMVS* against *BLS*, *New_K-OPT* and *BLS-RLE* on a broad range of real-world massive graphs, which can be categorized into 12 graph classes. The experimental results demonstrate that our *HSMVS* algorithm significantly outperforms *BLS*, *New_K-OPT* and *BLS-RLE* on a large number of real-world massive graphs with regards to both the best solution quality and the average solution quality, indicating that the superiority of *HSMVS* on solving real-world massive graphs. Furthermore, we conduct more empirical evaluations to confirm the effectiveness of the approximate best selection component and the random walk component. The related empirical results show that *HSMVS* generally performs much better than its all alternative versions on most real-world massive graphs, and thus indicates that the approximate best selection component and the random walk component make contributions to *HSMVS*.

We note that *HSMVS* is simple yet efficient. In this sense, *HSMVS* is able to serve as a good algorithmic framework, and more improved algorithms could be proposed and implemented on the top of it. For future work, to further improve the computational performance of MVS heuristic search algorithms, we would like to combine *HSMVS* with other algorithmic strategies proposed to handle other combinatorial problems, such as configuration checking (*Cai & Su, 2013*), weighting techniques (*Cai, Lin & Su, 2015*) and probability distribution (*Balint & Fröhlich, 2010*). We would also like to utilize powerful

automatic configuration tools (*Hutter et al., 2009*; *Hutter, Hoos & Leyton-Brown, 2011*) to improve the performance of *HSMVS*.

### Funding
This work was supported by the National Key Research and Development Program of China under Grant 2023YFB3307503, by the National Natural Science Foundation of China under Grant 62202025, by CCF-Huawei Populus Grove Fund under Grant CCF-HuaweiSY202311, and by the Frontier Cross Fund Project of Beihang University. There was no additional external funding received for this study. The funders had no role in study design, data collection and analysis, decision to publish, or preparation of the manuscript.

### Grant Disclosures
The following grant information was disclosed by the authors:
The National Key Research and Development Program of China: 2023YFB3307503.
The National Natural Science Foundation of China: 62202025.
CCF-Huawei Populus Grove Fund: CCF-HuaweiSY202311.
The Frontier Cross Fund Project of Beihang University.

### Competing Interests
The authors declare there are no competing interests.

### Author Contributions
- Chuan Luo conceived and designed the experiments, performed the experiments, analyzed the data, performed the computation work, prepared figures and/or tables, authored or reviewed drafts of the article, and approved the final draft.
- Shanyu Guo performed the experiments, analyzed the data, prepared figures and/or tables, and approved the final draft.

### Data Availability
The source code of HSMVS is available in the Supplementary File.

### Supplemental Information
Supplemental information for this article can be found online at http://dx.doi.org/10.7717/peerj-cs.2013#supplemental-information.

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
