# Peer review of "HSMVS: heuristic search for minimum vertex separator on massive graphs"

_PeerJ Computer Science, doi:10.7717/peerj-cs.2013_

## Round 0.1 · original submission · Major Revisions

I would encourage you to include an example as suggested by the reviewer and explain certain terminology. The paper overall appears to be a very good shape. the suggested modifications will bring it to the right level.

Please also document the changes you made so that reviewers can quickly verify them.

Looking forward to the revised version.

Reviewer 1 ·

Basic reporting

In this work, the authors present an effective MVS heuristic search algorithm called HSMVS. Compared to the existing, effective MVS heuristic search algorithm BLS, which incorporates several sophisticated heuristics and is involved in a number of parameters, the proposed HSMVS algorithm concentrates on only one simple yet effective heuristic and introduces fewer parameters.

To demonstrate the effectiveness of the HSMVS algorithm, the authors conducted extensive experiments to compare HSMVS against BLS and New K-OPT on a broad range of real-world massive graphs, which can be categorized into 12 graph classes. The experimental results demonstrate that our HSMVS algorithm significantly outperforms BLS and New K-OPT on a large number of real-world massive graphs with regards to both the best solution quality and the average solution quality, indicating that the superiority of HSMVS on solving real-world massive graphs.

Experimental design

Some of the positive aspects of the paper are as follows:
• The paper addresses a very important problem in graph theory that is central to several application domains and has wide applicability.
• The paper is well written, easy to read and has done a thorough investigation of 139 available large-scale networks and other heuristic algorithms that are available and well known in literature.
• The numerical and comparative analysis done by the authors in the paper seem to be quite rigorous.

Validity of the findings

Some of the comments on the paper are as follows:
• The authors didn’t ever define in the paper what does “balanced components” in the graph mean. They should define it mathematically so that it is precisely clear to the readers.
• It will be great if the authors provide a generalized observation and systematic explanation and intuition behind the impact of score_A and score_B based on the movement of vertices from one set to the other.
• Can the authors also explain the logical reasoning with a figure behind the “greedy mode” heuristic. That will aid the readers of the paper.
• The HSMVS algorithm sometimes take more time to converge to a better solution as compared to the BLS and New_K_Opt heuristic.
• It also sometimes achieves inferior solutions as compared to the other two heuristics.

Additional comments

• Are there any algorithmic insights that the authors can provide based on the topography of each graph (each graph can have a varying degree distribution and connectivity patterns). Is there any overall insights that the authors can provide that are derived out of those graph properties. That will help to generalize their solution to even broader class of graphs.

Cite this review as

Reviewer 2 ·

Basic reporting

Starting from the abstract, the authors mentioned “disconnects a graph into two nearly balanced components” several times. Here, the term “nearly balanced components” is not clearly defined. Based on my understanding, The conventional general “vertex separator” or “Minimal vertex separator” concepts do not have relevant consideration. How to evaluate if the two parts are balanced? Moreover, how does this be considered in the heuristic algorithm?

In the related work part, is there any recent MVS work? Many papers are cited in the 3rd paragraph of the introduction but most of the work that is directly related to MVS is not the latest work. Please cite more latest proposed MVS work.

The authors presented the algorithm with the pseudo-code, which is very clear. But I think the presentation could be better with an example, like, giving a small graph with the change of the set A, B and C. This will give readers a better understanding of the graph/topology perspective. Also, the same suggestion is to the subfunction “move” in section 5.2 and the algorithm “Modify_Solution”.

In line 305 (section 6.3), the term “best solution quality” suddenly appears without a definition. Backtrack to line 156 in section 4, the solution quality is mentioned here, but there is no explanation as well.

Two competitors are old. BLS is proposed in 2013 and New_K-OPT is proposed in 2015. Is there any recently proposed algorithm? If so, please add one more latest competitor.

Another very important parameter of the algorithm is the value of “b”, which is not investigated here. How will the algorithm perform with different values of b? Will the algorithm still be better than the two baseline algorithms? If b is getting larger, will the problem get easier or harder? And will the algorithm come up with correspondence results? Like, for easier problems with good results and harder problems with longer time.

The authors presented the performance of HSMVS, BLS, and New_K-OPT in different networks. I think the result could be explained more as follows:
1. The results of different types of networks are shown, but the features of networks are not mentioned. I think biological networks, collaboration networks, Facebook networks, infrastructure networks, etc., should have different features in terms of topology, size, number of bridge vertex, etc. The authors tried to cover various networks, but the demonstration of these presentative networks could make them more reasonable.
2. We can see HSMVS is better than the two baseline algorithms, but the improvements are not similar. Some times are much better but sometimes very close. I think the reason for this should be explained, instead of just showing the gap.
3. Meanwhile, if the improvements are not similar among different networks, I think that means the algorithm may be sensitive to some features of the networks. Please bring more explanations here. Currently, the performance results of different networks are separate.

Experimental design

No comment

Validity of the findings

No comment

Cite this review as

---

## Round 0.2 · accepted · Accept

Thanks for your submission and patience while we waited for the reviewers' comments. Both the reviewers are now happy and recommend acceptance. They are complimentary of you being responsive to their earlier comment. Thanks for your efforts.

Reviewer 1 ·

Basic reporting

In this work, the authors present an effective MVS heuristic search algorithm called HSMVS. Compared to the existing, effective MVS heuristic search algorithm BLS, which incorporates several sophisticated heuristics and is involved in a number of parameters, the proposed HSMVS algorithm concentrates on only one simple yet effective heuristic and introduces fewer parameters.

The authors have responded to all the questions raised by the reviewers in their response letter and have appropriately also adapted the main text of the paper to make it more understandable to the readers.
The current revised manuscript now reads more clear and addresses all the concerns and questions.

Experimental design

The authors have added more elaborate changes in the algorithm description and how it's working. They have also added substantial changes in the draft to make it more legible to the readers.

Validity of the findings

The explanations provided by the authors in the revised draft and in the attached letter makes their claims more clear and easy to validate with some examples.

Cite this review as

Reviewer 2 ·

Basic reporting

This is the second review of the paper "HSMVS: Heuristic Search for Minimum 2 Vertex Separator on Massive Graphs", in which I acted as Reviewer 2 in the first round.

The authors have adequately addressed my comments and requests. I would like to acknowledge the effort of providing a new demonstration of the "balanced components", adding new examples with new figures, and the latest references.

I also appreciate the clear explanation of the simulation setting issues and the newly added results in the tables.

From my perspective, the paper is now acceptable for publication.

Experimental design

The authors have adequately addressed my comments and requests.

Validity of the findings

The authors have adequately addressed my comments and requests.

Additional comments

The authors have adequately addressed my comments and requests.

Cite this review as